# The Farmed Atlantic Salmon (*Salmo salar*) Skin–Mucus Proteome and Its Nutrient Potential for the Resident Bacterial Community

**DOI:** 10.3390/genes10070515

**Published:** 2019-07-07

**Authors:** Giusi Minniti, Simen Rød Sandve, János Tamás Padra, Live Heldal Hagen, Sara Lindén, Phillip B. Pope, Magnus Ø. Arntzen, Gustav Vaaje-Kolstad

**Affiliations:** 1Faculty of Chemistry, Biotechnology and Food Science, Norwegian University of Life Sciences (NMBU), NO-1432 Ås, Norway; 2Faculty of Biosciences, Norwegian University of Life Sciences (NMBU), NO-1432 Ås, Norway; 3Department of Medical Biochemistry and Cell Biology, Sahlgrenska Academy, University of Gothenburg, SE-405 30 Gothenburg, Sweden

**Keywords:** teleost, *Salmo salar*, skin–mucus, microbiome, proteome, aquaculture

## Abstract

Norway is the largest producer and exporter of farmed Atlantic salmon (*Salmo salar*) worldwide. Skin disorders correlated with bacterial infections represent an important challenge for fish farmers due to the economic losses caused. Little is known about this topic, thus studying the skin–mucus of *Salmo salar* and its bacterial community depict a step forward in understanding fish welfare in aquaculture. In this study, we used label free quantitative mass spectrometry to investigate the skin–mucus proteins associated with both Atlantic salmon and bacteria. In particular, the microbial temporal proteome dynamics during nine days of mucus incubation with sterilized seawater was investigated, in order to evaluate their capacity to utilize mucus components for growth in this environment. At the start of the incubation period, the largest proportion of proteins (~99%) belonged to the salmon and many of these proteins were assigned to protecting functions, confirming the defensive role of mucus. On the contrary, after nine days of incubation, most of the proteins detected were assigned to bacteria, mainly to the genera Vibrio and Pseudoalteromonas. Most of the predicted secreted proteins were affiliated with transport and metabolic processes. In particular, a large abundance and variety of bacterial proteases were observed, highlighting the capacity of bacteria to degrade the skin–mucus proteins of Atlantic salmon.

## 1. Introduction

Mucus is a complex, viscous, hydrated secretion present at the interface between the epithelial surface and its external environment. The dominant constituents of mucus are mucins, a family of highly glycosylated proteins that form large macromolecular networks through intermolecular disulfide bonds [1]. In addition to mucins, mucus contains salts, lipids, and a variety of proteins, many related to defense functions like lysozyme, defensins, immunoglobulins, growth factors, and trefoil factors [2]. In most of the mucosal surfaces, mucus has a protective function, providing a semipermeable but robust barrier that prevents unwanted chemical compounds, pathogens, and parasites from reaching the epithelial cell surfaces [3]. For instance, it is well known that mucins play a key role in accommodating a resident bacterial community and limiting adhesion of pathogens [1]. Specifically, it seems that changes in mucin glycan composition and possibly release of mucins with altered binding properties enable removal of microbes from different mucosal surfaces in order to prevent infections [4,5,6,7]. However, some pathogenic microorganisms have developed efficient strategies to overcome the mucosal barriers [8]. For instance, mammalian intestinal pathogens seem to bypass the intestinal mucus layers by using mucin-specific proteases [9,10]. 

It is well known that the skin of teleost fish is covered by mucus, which is involved in osmo- and ion regulation [11], gas exchange [12], reduction of fluid friction [13], and defense [14]. As all the mucosal surfaces, the fish skin–mucus harbors a complex microbial community (symbiotic and/or opportunistic) [15,16] and a correct balance between them seems to be crucial to ensure fish health [17]. In a study by Padra and colleagues [18], it was observed that the mucins present on the skin–mucus of salmon were characterized by shorter glycans and lower level of N-acetylneuraminic acid, compared to the mucins of the salmon intestinal regions, which manifested itself in the form of low adhesion of the fish pathogen *Aeromonas salmonicida*. This suggests that the skin–mucus composition may play an important role in controlling the microbial community living on the fish, in accordance with the abovementioned studies of other mucosal surfaces. Furthermore, Sveen et al. recently showed that expression of mucin encoding genes in the skin of Atlantic salmon decreased upon handling stress, but increased in the time period following long term stress [19], highlighting the fact that environmental factors have influence on mucus composition and in turn on the skin–mucus microbiome composition. However, little is known regarding how bacteria can attach to and bypass the mucus and promote infection. 

The biochemical constituents of fish skin–mucus has been studied by many, using primarily enzyme and immunological assays [20,21,22,23,24,25,26,27,28,29,30] and proteomics [31,32,33,34,35,36,37,38] (see also Brinchmann et al. (2016) [39] and Ángeles Esteban (2012) [40] for comprehensive reviews). As already noted, many of the free proteins are related to protective roles against infection. Commonly identified antimicrobial proteins are lysozymes [41], histone fragments [42], antibodies [43], lectins [44], complement components, alkaline phosphatase, and proteases [45]. In addition, most of the proteomic studies report the findings of proteins related to cellular function such as actin, collagen, ribosome subunits, and proteins related to metabolic pathways. It is not known whether such proteins originate from dead or living cells present in the mucus or if they are truly secreted with a specific biochemical function. In addition, as mentioned previously the presence of a bacterial community in the fish skin–mucus also contributes to the presence of bacterial proteins secreted for the purpose of protection, adhesion, feeding, proliferation or other life sustaining functions. However, only one study [33] has reported identification of bacterial proteins in fish mucus. 

In the current study, label-free quantitative mass spectrometry was used to investigate the exoproteome of the *Salmo salar* skin–mucus and its resident bacteria. The microbial community able to utilize mucus components as nutrients was analyzed by 16S rRNA gene amplicon sequencing, accompanied with temporal exoproteome data, yielding an opportunity of not only identifying the bacteria present in the skin–mucus, but also for functional analysis and determination of the microbial metabolic strategies. Finally, the exoproteome associated with the skin–mucus of *Salmo salar* was also analyzed in order to establish its composition. 

## 2. Materials and Methods

### 2.1. Fish and Sampling Procedure

Eight farmed *Salmo salar* (average weight 300 g) from the Norwegian Institute of Water Research (NIVA), Drøbak, Norway, were randomly sampled, immediately killed by a blow to the head and transferred to an autoclaved plastic bag and stored on ice. Of these fish, three (named F1, F2 and F3) were used for skin–mucus label-free quantitative proteomic analysis, whereas the remaining five (named F4, F5, F6, F7, and F8) were selected to estimate the skin–mucus mucin concentration (see below for method). The net used for sampling the fish was sterilized with 70% EtOH between each round. The tank biomass was approximately 1 kg/m^3^ and the tank was supplied with seawater from 60 m depth in the Oslo Fjord with salinity of 34.7 PSU and a temperature of 7.6 °C. Fish were fed with Skretting Spirit 4.5 mm pellets at ~0.5% fish weight per day. Mucus was sampled in two steps: first, by draining the mucus accumulated in the sterilized plastic bag into a 1.5 mL plastic test tube, followed by gentle collection of the skin–mucus by a rounded plastic spatula and transfer of this mucus to the same test tube. Sampled mucus (~3 mL) from each of the three individuals (F1, F2, and F3) was split into three separate sterile culture tubes (1 mL per tube), mixed with 9 mL sterilized seawater and gently vortexed, yielding three technical replicates for each mucus sample. All samples were incubated for 9 days at 10 °C on a shaker (200 rpm). A total volume of 750 µL was sampled from each technical replicate after 0 (i.e., immediately after dilution of the mucus and the subsequence mixing), 2, 5, and 9 days. All samples were divided in two: 100 µL for 16S rRNA gene sequencing analysis and 650 µL for proteomics analysis. The latter samples were centrifuged at 5.500 g for 10 minutes. The supernatants and pellets were stored at −20 °C until needed for proteomics- or 16S rRNA sequencing analysis, respectively (see below for details). The mucus samples (not diluted) from the remaining four fishes (F4, F5, F6 and F7) were stored at −20 °C until mucin quantification. Finally, as mentioned above, one additional farmed salmon (F8) was successively collected from the same farming facility for the sake of conducting a supplementing skin–mucus microbe growth experiment. This fish (~3000 g) had been exposed to the same conditions as stated above. Skin–mucus collection and cultivation was also performed as stated above, but in this case, the mucus was not diluted in sterile seawater, in order to maintain as high mucin concentration in the sample as possible. Samples were collected at 0, 12, 24, and 48 hours and stored at −20 °C until mucin quantification (see below).

### 2.2. Sample Preparation for 16S rRNA Gene Sequencing

DNA was extracted from bacteria grown in the diluted mucus preparation after incubation for 9 days at 10 °C at 200 rpm using the DNeasy tissue kit (Qiagen, Hilden, Germany), following the protocol for Gram-positive bacteria with some modifications. Achromopeptidase (1h incubation at 37 °C) was utilized in the first step of the DNA extraction procedure in order to ensure lysis of Gram-positive bacteria [46]. Proteinase K (40 μL) and ATL buffer (180 μL) were added to the samples, and tubes were incubated at 55 °C for 1h. Successively, AL buffer (200 µL) was used as last lysis step (incubation at 70 °C for 10 minutes). The manufacture’s protocol was followed during the remaining steps. The extracted DNA was stored at −20 °C. Sample preparation for 16S rRNA sequencing analysis by MiSeq was performed according the Illumina guide (16S Metagenomic Sequencing Library Preparation, Part. 15,044,223 Rev A). For this analysis, the primer set targeting the V3-V4 hypervariable regions, Pro341F (5’-CCTA CGGGNBGCASCAG-3’) and Pro805R (5’-GACTACNVGGGT ATCTAATCC-3’) [47] and the polymerase iProof High-Fidelity (Bio-Rad, Hercules, CA, USA) were used to amplify the 16S rRNA genes. The PCR conditions used were as follows: initial denaturation at 95 °C for 3 minutes, followed by 30 cycles of 95 °C for 30 seconds, 55 °C for 30 seconds, 72 °C for 30 seconds, and concluded by a final extension at 72 °C for 5 minutes. The quality of the PCR products was validated by agarose gel electrophoresis and quantified by Qubit dsDNA HS Assay (Invitrogen, Carlsbad, CA, USA). MiSeq (Illumina, San Diego, CA, USA) was used to sequence the mucus samples at Day 9. 

### 2.3. Analysis of 16s rRNA Gene Sequencing Data

Analysis of the data was accomplished using the Usearch v.8.1861 [48] within the QIIME v1.8.0 [49] pipeline. First, paired reads were merged, quality filtered (E_max = 1) and singletons were removed. Reads were trimmed to 430 nucleotides. Subsequently, the command cluster_otus was used to identify and remove chimeric sequences from the database and to cluster the Operational Taxonomy Units (OTUs), using the UPARSE-OTU algorithm (97%). Finally, taxonomy was assigned employing the UTAX algorithm [48] with the full-length RDP training set (utax_rdp_16s_trainset15), enabling the construction of the OTU table. The data was further normalized by rarefying to an even sample depth (28.693 sequences/sample) in order to remove sample heterogeneity. Alpha diversity was represented by rarefaction curves through two indices: Chao1 and observed-species. In order to focus on the most abundant taxa, the OTUs represented by less than 0.2% of the total reads were filtered out, although they were displayed in the taxonomy plot (Figure 1) as “Others”. The most abundant OTUs (>0.2%) were utilized to create a community database used for the proteomic analysis. 

### 2.4. Sample Preparation and Proteomic Analysis

The mixture of mucus and seawater was centrifuged at 5.500 × *g* for 10 minutes and the supernatant was collected and stored at −20 °C until use. Protein precipitation was performed by adding 50% (*v*/*v*) Trichloroacetic acid to a final concentration of 10% (*v*/*v*), followed by incubation for 1 h at 4 °C. Precipitated proteins were centrifuged for 15 min at 15.000 × *g*, washed one time with 0.01 M HCl/90% (*v*/*v*) acetone and centrifuged again for 15 min at 15.000 × *g*. Subsequently, pellets were resuspended in SDS sample buffer and separated by SDS-PAGE using the AnykD Mini-PROTEAN gel (Bio-Rad Laboratories, Hercules, CA, USA) at 240 V for 12 minutes and stained using Coomassie Brilliant Blue R250. Each gel lane was cut into five bands and destained two times using 25 mM ammonium bicarbonate in 50% (*v*/*v*) acetonitrile. The proteins were reduced and alkylated at 25 °C using 10 mM DTT and 55 mM iodacetamide, respectively, with an incubation time of 30 minutes in each step. Afterwards, proteins were digested overnight with 0.1 µg trypsin (Promega, Mannheim, Germany) in 25 mM ammonium bicarbonate at 37 °C as previously described [50]. Prior to mass spectrometry, peptides were desalted using C18 ZipTips (Merck Millipore, Darmstadt, Germany), dried under vacuum (45 °C) (Concentrator plus, Eppendorf, Denmark), and dissolved in 2% (*v*/*v*) acetonitrile and 0.1% (*v*/*v*) tri-fluoro acetic acid. Peptides were analyzed using a nanoHPLC-MS/MS system consisting of a Dionex Ultimate 3000 UHPLC (Thermo Scientific, Bremen, Germany) connected to a Q-Exactive hybrid quadrupole-orbitrap mass spectrometer (Thermo Scientific, Bremen, Germany) equipped with a nano-electrospray ion source. Samples were loaded onto a trap column (Acclaim PepMap100, C_18_, 5 µm, 100 Å, 300 µm ID × 5 mm, Thermo Scientific) and back flushed onto a 50 cm analytical column (Acclaim PepMap RSLC C_18_, 2 µm, 100 Å, 75 µm ID, Thermo Scientific). At the beginning, the columns were in 96% solution A [0.1% (*v*/*v*) formic acid], 4% solution B (80% (*v*/*v*) acetonitrile, 0.1% (*v*/*v*) formic acid). Peptides were eluted using a 90-min gradient developing from 4% to 13% (*v*/*v*) solution B in 2 minutes, 13% to 45% (*v*/*v*) B in 70 minutes and finally to 55% B in 5 minutes before the washing phase at 90% B. The flow rate was constant at 300 nL/min. In order to isolate and fragment the 10 most intense peptide precursor ions at any given time throughout the chromatographic elution, the mass spectrometer was operated in data-dependent mode (DDA) to switch automatically between orbitrap-MS and higher-energy collisional dissociation (HCD) orbitrap-MS/MS acquisition. The selected precursor ions were then excluded for repeated fragmentation for 20 seconds. The resolution was set to R = 70.000 and R = 35.000 for MS and MS/MS, respectively. For optimal acquisition of MS/MS spectra, automatic gain control (AGC) target values were set to 50,000 charges and a maximum injection time of 128 milliseconds.

### 2.5. Analysis of Proteomic Data

A community database was generated by extracting and concatenating UniProt or NCBI protein entries for the 18 genera detected in the 16S rRNA analysis (Figure 1). In detail, 16 genera were extracted from UniProt (SwissProt + TrEMBL, release 2017_01) (Bacteriovorax, Colwellia, Halomonas, Litoreibacter, Lysobacter, Marinomonas, Methylobacterium, Neptuniibacter, Phaeobacter, Polaribacter, Pseudoalteromonas, Ralstonia, Shewanella, Sphingomonas, Sulfitobacter, and Vibrio) and two were extracted from NCBI (Olleya and Rubritalea). In addition, UniProt entries for Salmo were added in the database for the detection of salmon proteins. In total, the database had 2.297.141 protein entries. MS raw files were analyzed using MaxQuant version 1.4.1.2 [51] and proteins were identified and quantified using the MaxLFQ algorithm [52]. In details, the data were searched against the abovementioned community database supplemented with common contaminants such as human keratin and bovine serum albumin. In addition, reversed sequences of all protein entries were concatenated to the database for estimation of false discovery rates. The tolerance levels for matching to the database was 6 ppm for MS and 20 ppm for MS/MS. Trypsin was used as digestion enzyme, and two missed cleavages were allowed. Carbamidomethylation of cysteine residues was set as a fixed modification and protein N-terminal acetylation and oxidation of methionines were allowed as variable modifications. The “match between runs” feature of MaxQuant, which enables identification transfer between samples based on accurate mass and retention time [52], was applied with a match time window of one minute and an alignment time window of 20 minutes. All identifications were filtered in order to achieve a protein false discovery rate (FDR) of 1% and further filtering were applied to include at least one unique peptide and at least two peptides in total. Proteins that could not be unambiguously assigned to one genus (2.5% of total protein identifications), i.e., contained only peptides with shared sequences across different genera, were omitted from further analysis. Regarding protein quantifications, the label-free quantification (LFQ) values reported by MaxQuant were used to calculate the protein abundances at the genus level. The LFQ values were summed for each genus and then log-transformed before developing the heat maps and sample comparisons. The LipoP 1.0 server [53] was used to predict the sub-cellular locations of the detected proteins nd dbCAN [54] was utilized for the prediction of carbohydrate-active enzymes. Finally, Gene Ontology (GO) [55] was used for functional annotation of the detected bacterial proteins and salmon skin–mucus proteins. The proteins were further categorized into groups of GO-terms using high-level GO-SLIMs. The GO-SLIMs, one for bacterial proteins and one for salmon proteins, were constructed using QuickGO [56] and based on the most prevalent GO terms in the dataset and with minimized hierarchical overlap in the GO structure.

### 2.6. Mapping of Salmo salar Mucus Proteins to RNAseq Data

The Atlantic salmon RNA-seq expression data samples from 15 tissues were obtained from NCBI SRA (PRJNA72713) [57]. Fastq files were adapter-trimmed before alignment to the Atlantic salmon genome (RefSeq assembly GCF_000233375.1) using STAR (v2.5.2a) [58]. We used HTSeq-count (v0.6.1p1) [59] to produce counts of uniquely aligned reads to each gene (RefSeq Annotation Release 100), and then calculated the relative gene expression levels as per kilobase of exon per million reads mapped after normalizing for the samples effective library size (see TMM method from edgeR user manual [60]).

All downstream analyses were carried out in R (version 3.4.1). For the relationship between protein amount and gene expression, a linear model was fitted using the lm function, and spearman correlation test was performed using log2 transformed values. The resampling test for skin-expression bias in the skin protein gene set (derived from mapping salmon proteins detected by proteomics to RefSeq genes), was constructed using the sample function in R without replacement. Number of resampled top expressed genes in skin was computed from 1000 resampled sets of 3158 genes from all protein coding genes in *Salmo salar*. Resampling *p*-value was calculated as followed: *p*-value = (r+1)/1,000, where r is the number of resampled datasets with equal or more genes having highest expression in skin compared to the genes encoding proteins detected in Atlantic salmon skin–mucus. Heatmap of skin expression levels of the 299 genes with the highest expression in skin was plotted using the pheatmap package. Expression values were scaled by rows and clustered using spearman correlation distance.

### 2.7. Mucin Quantification

Atlantic salmon mucin samples were treated with protease inhibitor (Sigma-Aldrich Co.) diluted 1:100, aliquoted and stored at −80 °C between experiments. A ten step serial dilution of pig gastric mucin (PGM, Sigma-Aldrich Co.) with known concentration and glycosylation was used as a standard for concentration determination in the range of 500 µg/mL and 1µg/mL. PVDF membranes (Merck Millipore Ltd.) with 0.45 µm pore size and blot papers (Bio-Rad Laboratories) were pretreated with methanol (Sigma-Aldrich Co.) prior to use and 100 µL sample (diluted 1:500, 1:250, and 1:125) and PGM were slot blotted on to the PVDF membranes. After the samples were sucked through, 100 µl dH_2_O were added to the wells to retain all sample material. The membrane was air-dried, dipped in methanol, and then dipped in 3% acetic acid. The membrane was stained with 1% Alcian blue 8GX (Sigma-Aldrich Co.)/3% Acetic acid (pH2.5) for 1h and then destained three times in methanol. The membrane was scanned with a Bio-Rad Gel Doc^TM^ EZ Imager and the intensity of the bands was quantified with the ImageJ software. The concentration of mucin samples was calculated from band intensities using PGM standard curves. The concentrations were corrected for the glycosylation differences between Atlantic salmon and PGM, i.e., the glycan chain length and the ratio of charged glycan structures [61]. The assay was reproduced three times.

### 2.8. Nucleotide Sequence Accession Numbers

Sequence data are available at NCBI Sequence Read Archive under accession number SRP123429.

### 2.9. Mass Spectrometry Proteomic Data

The mass spectrometry proteomic data have been deposited to the ProteomeXchange Consortium via the PRIDE [62] partner repository with the dataset identifier PXD008838.

## 3. Results

### 3.1. 16S rRNA Gene Sequencing Analysis of Bacteria Utilizing Salmon Skin–Mucus as Nutrient Source 

In order to obtain an overview of the bacteria able to proliferate in the *Salmo salar* skin–mucus, 16S rRNA gene analysis was used to investigate the bacterial community established after incubating mucus in sterile seawater for nine days at 10 °C. After filtering low-quality reads and normalizing to an even sample depth, a total of 258,237 reads were identified from nine samples (three technical replicates from fish F1, F2, and F3), which could be assigned to 297 OTUs. The rarefaction curves (Appendix A) suggested that a sufficient sequencing coverage was used to study the microbial diversity that was present within the samples. In order to focus on the most abundant taxa, only the OTUs ≥ 0.2% of the total number of sequences were utilized to create a taxonomy plot (Figure 1) and to construct a community database for proteomics. The resulting data set showed a total number of 249,070 reads distributed among 23 OTUs assigned to 18 genera. The microbial community amongst the three biological replicates showed *Vibrio* as the most abundant genus in samples F1 and F2. Rubritalea and Olleya were the second most abundant genera. Both F1 and F2 exhibited a consistent bacterial profile in the three technical replicates. On the contrary, the bacterial community identified in F3 deviated substantially from F1 and F2, as Vibrio, Rubritalea, and Olleya were present in low amounts, while Bacteriovorax and Ralstonia were the dominating genera. A large technical variation was observed among the three replicates in F3. 

### 3.2. Temporal Quantitative Proteomic Analysis of the Salmon Skin Microbiome

Label-free quantitative proteomics was utilized to monitor the temporal dynamics of the skin–mucus microbial exoproteome. A total number of 4563 bacterial proteins were detected in the analysis over the nine days of incubation (Table 1 and Figure 2). Medium to high reproducibility of quantification was observed between the biological replicates at the different experimental time points (Appendix A, Pearson correlation R = 0.39–0.95). The majority of the proteins detected in the analysis were assigned to Vibrio and Pseudoalteromonas genera, with 2017 and 1621 proteins identified, respectively (Table 1). Shewanella, Neptuniibacter, and Marinomonas were also represented with 519, 154 and 146 proteins, respectively. Furthermore, 13 additional bacterial genera were detected with low protein counts. In total, 996 bacterial proteins (22%) were predicted to be actively secreted (Table 1). 

The taxonomic proportions changed markedly during the nine days of incubation of salmon skin–mucus in sterilized seawater (Figure 2). At day 0, 99.2% of the proteins belonged to *Salmo salar* while 0.8% of the proteins were from bacterial origin (Vibrio 0.6%, Pseudoalteromonas 0.14%, Shewanella 0.03%, and Olleya 0.03%). After five days, the *Salmo salar* proteins were substantially reduced and after nine days bacterial proteins dominated the mucus exoproteome. Displaying the protein abundances as a heat map (Figure 3), we observed that Pseudoalteromonas, Shewanella, and Vibrio showed a strong increase of their abundances over time, as did Neptuniibacter, Marinomonas, Halomonas, and Colwellia, but in lower amounts. Ralstonia and *Sulfitobacter* were only detectable at day 2 and onwards, whereas Sphingomonas and Olleya showed a consistent abundance during all experimental time points with minor fluctuation in their abundances. Methylobacterium showed a decrease in abundance over time. Finally, Bacteriovorax, Polaribacter, Lysobacter, Phaeobacter, and Rubritalea were identified, but in few replicates and time points. The remaining genus, Litoreibacter, could not be identified at all during the analyzed time period. 

In order to get an overview of the functional involvement of the expressed bacterial exoproteome, we used a gene ontology (GO) approach. The resulting table revealed a variety of processes represented in the exoproteome such as regulation, transport, metabolism, pathogenesis, cellular processes and many more (Appendix A). Since the functionally relevant proteins in an exoproteome are those actively secreted by the bacteria, the identified proteins were also sorted based on the presence of a predicted leader peptide indicating secretion. When focusing on predicted secreted proteins from the dominant Vibrio genus, GO clustering showed proteins mainly associated with transport and metabolic processes (Figure 4, panels A and B). In particular, proteins assigned to siderophore transport and proteolytic activity were abundant (Figure 4, panels C and D). Among the siderophore transport proteins, nine proteins stand out; the Vitamin B12 transporter BtuB (ADA1C3IP31), ferrichrome iron receptor (ADA1C3IT79), ligand-gated channel protein (ADA1B9QID6), FhuE receptor (ADAA1C3J0C3), ferrichrome–iron receptor (A0A1C3IR55), ligand-gated channel protein (A0A1D2YG10), putative Fe-regulated protein B (A3UTN9), putative ferrichrome–iron receptor (A3UUWW6), and outer membrane receptor protein (A3UTS4). The expression of ion/siderophore transport proteins seemed to be persistent in all the samples from day 2, although some inter-fish fluctuations occur (Figure 4, panel C). In addition, several types of proteases from Vibrio were detected, spanning many of the protease families as defined by the MEROPS database [63] (e.g., peptidase M4, peptidase M16, alkaline serine protease, and aminopeptidases). The abundances of the individual proteases also varied within samples and sampling days (Figure 4; panel D). Overall, the most abundant proteases displayed were Peptidase M4 (A0A0P6Z347), Immune Inhibitor A (A0A1C3IQR9 and A0A1C3JD33) and Periplasmic protease (A3UZH0). The second most dominant bacterium detected in the secretome analysis was the genus Pseudoalteromonas (Table 1). Similar to Vibro, Pseudoalteromonas also showed high expression of proteins associated with metabolic processes (proteolysis) and transport (siderophore transport) (Appendix A, panels A–D) indicating the necessity of such enzymes to survive and proliferate in the *Salmo salar* mucus. 

### 3.3. Host Proteins Detected in the Salmo salar Skin–mucus

Quantitative proteomic analysis of the salmon skin–mucus exoproteome exhibited a total of 3583 proteins associated with Atlantic salmon (Table 1). As expected, the *Salmo salar* proteins were most abundant at the start of the incubation period (2 days), while the remaining experimental time points were dominated by bacterial proteins (Figure 2 and Figure 3). Similar to the bacterial, the salmon exoproteome was categorized according to biological function using a Gene Ontology (GO) approach. The GO analysis revealed that the majority of the proteins were involved in biological regulation processes, transport processes, metabolic processes, cellular processes and multiorganism processes (Appendix A), typically related to roles in the cytoplasm of cells. Indeed, only 15% of proteins identified were predicted to be actively secreted (Table 1 and Appendix A), indicating cell lysis or secretion by nonclassical mechanisms. Some of the hallmarks of mucosal immune responses were identified, such as multiple lysozyme, lectins, calmodulin, galectins, histones, ribosomal proteins, and complement-related proteins. Structural proteins such as actin, keratin, tubulin, cofilin-2 and filamin-A (Appendix A) were also observed. Interestingly, six of the ten chitinases encoded in the *Salmo salar* genome were detected (accession numbers A0A1S3L8D8, A0A1S3L8T9, A0A1S3LA77, A0A1S3MFN1, A0A1S3N6L1, A0A1S3P6P2, and B5DG80), which may confer protection against chitin containing pathogens (e.g., fungi) or parasites (e.g., sea lice).

In order to investigate if the salmon proteins identified in the day 0 samples are likely formed (and secreted) from active/living, skin cells, the *Salmo salar* exoproteome was mapped against an existing RNAseq panel from 15 tissues. Of the 3583 proteins identified in the exoproteome, 3158 could be uniquely mapped to salmon genes (hereafter referred to as “skin protein genes”) (Appendix A). There was a significant positive correlation between the protein abundance and steady state expression level of skin protein genes (Appendix A, panel A, Pearson correlation *p*-value = 4.59^−75^). A total number of 299 skin protein genes (9%) showed the highest gene expression level in skin in the gene expression tissue panel (Appendix A, panel C), which is a significant enrichment of genes with skin-biased expression (Appendix A, panel B) (resampling test *p*-value < 0.001). 

### 3.4. Quantification of Mucins

The concentration of mucin was estimated in four mucus samples, showing concentrations ranging from 9.9 to 22.2 mg/mL (Table 2). For one sample (F8), the depletion of mucin over time was also monitored in three technical replicates during bacterial growth in undiluted mucus. Over 48 hours the bacteria present had consumed more than 50% of the mucin available (Table 2).

## 4. Discussion

### 4.1. Skin Mucus Can Be Used by Microorganisms as a Source of Nutrients

In order to explore the salmon skin–mucus microbiome and its exoproteome, 16S rRNA sequencing analysis was used to determine the bacterial community and to guide construction of the community database. Unfortunately, only mucus samples incubated extensively (nine days) in sterile sea water to allow bacterial growth on mucus components, yielded DNA of sufficient quality to allow 16S rRNA amplicon analysis. This was not unexpected, as the farmed Atlantic salmon skin–mucus has been indicated to contain low bacterial biomass [64]. Additionally, it may be that the multiple host nucleases detected in the mucus (Appendix A) has had influence on the quality of the extracted DNA. Lacking a 16S rRNA based database for the “fresh” mucus, the proteomics analysis was performed in two ways: (1) using a database generated from the 16S rRNA analysis obtained at day 9 of mucus incubation and (2) using a comprehensive public repository-derived database containing the whole UniProt bacterial section. Both approaches yielded similar results (the latter data not shown), thus it is likely that what is detected after nine days of mucus incubation reflects the dominant species at day 0 (as is indicated by the proteomic data). 

The 16S rRNA community data obtained from analysis of the mucus incubated at 10 °C for nine days displayed, *Vibrio* as the dominant genus detected (Figure 1). *Vibrio sp.* are Gram-negative bacteria that live mainly in freshwater, seawater, and sediment [65], although they also have been recovered from the skin of freshwater and seawater fish [66]. This genus is often associated with pathogenesis in fish [67,68,69], highlighting its capacity to proliferate on mucosal surfaces, providing a possible explanation for their dominance in the exoproteome both before and after mucus incubation (Figure 2).

Blast comparison of the sequences assigned to Vibrio OTUs against the NCBI nucleotide database showed high percentage of identity (99%) with *Vibrio splendidus* and *Vibrio tapetis* species, which are well documented fish pathogens [70,71,72]. Despite the dominance of *Vibrio sp.*, variations in the bacterial profile was observed among the three sampled fish (Figure 1), which may be correlated with a different bacterial community present in each sample. Variability has been reported among species, individuals and different body parts [16,73,74]. Moreover, the bacterial diversity identified in the mucus after nine days of incubation might also be influenced by nonresident bacteria (e.g., water microbiome) that adhere to the mucus components [64] and grew during the nine days of incubation. 

As stated above, the 18 bacterial genera obtained from the 16S rRNA sequencing analysis were used to generate a community database, which was utilized for determining the exoproteome associated with the microbiome proliferating in mucus and to investigate its temporal dynamics. Due to the lack of a sample-specific metagenome database in our proteomic analysis, we used the second best approach, i.e., a sample-specific filtering of protein sequences in public repositories based on the 16S rRNA analysis, a strategy also suggested in other studies [75,76]. When applying this 16S rRNA based sample-specific filtering, we could reduce the public repository-derived database from 58M (whole UniProt bacterial section) to 2.2 M and, by this, increase the sensitivity of the analysis. However, we acknowledge that our approach may fail to include sample-specific strain variations in the microbial community as well as running the risk of being incomplete in terms of community structure; the results must be interpreted in light of this. Using this database, we achieved an overall ID-rate of 42%, i.e., 42% of all MS/MS spectra could be assigned to a protein; this high number indicates that our database has good quality in terms of completeness. 

In general, we detected proteins involved in a variety of biological processes (Appendix A), although representatives of metabolic (proteases) and transport functions (iron transport) dominated. From a proteomic study conducted by Ræder et al on the response of *Vibrio salmonicida* (now known as *Aliivibrio salmonicida*) to mucus exposure, enhanced levels of flagellar proteins, heat shock proteins, chaperonins, and a variety of peroxidases were observed and suggested to enable survival of the bacterium in the mucus of salmon [36]. Indeed, proteins involved in motility (flagellin), oxidative stress response (peroxidases), and general stress response (heat shock proteins and chaperones) were observed (Appendix A). The presence of flagellin is documented to be needed by pathogenic bacteria to colonize different environmental niches [77,78,79]. Thus, it may be important for bacterial colonization of viscous environments like mucus. Despite the identification of the important mucus growth factors mentioned above, the dominant group of proteins in the bacterial exoproteome were related to proteolysis and iron transport (see below for further discussion). It should be noted that as bacteria are growing, cell lysis will ultimately occur giving rise to a substantial amount of cytoplasmic proteins in the exoproteome [80]. Thus, the detection of multiple intracellular proteins in the present study may indicate an active defense response of the *Salmo salar* mucus (several antimicrobial proteins from the salmon were identified; see discussion further below), bacterial proteins are secreted by unconventional secretion mechanisms or simply are the result of a natural cell lysis. 

### 4.2. Proteases and Siderophore Transporters Dominate the Bacterial Exoproteome

The majority of proteins identified in the exoproteome were mapped to Vibrio and Pseudoalteromonas. Vibrio was the most dominant genus observed from both 16S rRNA gene sequencing and proteomic analysis (Table 1 and Figure 1 and Figure 2), which was followed by the genus Pseudoalteromonas according to our proteomic dataset (Table 1 and Figure 2). For this reason, a greater emphasis will be put on these genera in the discussion below. In general, bacterial proteins were mostly identified after five to nine days of mucus incubation (Figure 2). Specifically, looking at the abundances of intracellular and secreted proteins by Vibrio and Pseudoalteromonas during the nine days of mucus incubations, higher protein expression was clearly observed over the last experimental time series (Figure 3). The increase of bacterial proteins was correlated with a decrease of salmon proteins (Figure 2 and Figure 3), inferring bacterial utilization of salmon proteins as a nutrient source. Indeed, the GO analysis showed a high number of secreted proteases from both *Vibrio* (Figure 4, panels B and D) and *Pseudoalteromonas* (Appendix A, panels B and D). In a study on the mollusk pathogens *Vibrio tapetis* and *Vibrio eastuarianus* secretomes, only few predicted extracellular enzymes were detected, these being proteases, lipases and chitinases, all suggested to be virulence factors. Many of the proteases identified in our analysis are known as virulence factors, such as vibriolysine [81], haemmagglutinin [82], and immune inhibitor A [83]. In addition, one of the most abundant proteases secreted by the genus Vibrio is the peptidase M4 (A0A0P6Z347) (Figure 4; panel D), which is a metallopeptidase. Different families of metallopeptidases are have been identified in pathogenesis [84,85,86]. Interestingly, chitinases were also identified in the current exoproteome (Figure 4 panel B and Appendix A). Although no obvious function can be proposed for chitinases in the skin mucus, it has been suggested that *Salmo salar* scales contain chitin [87], the natural substrate for these enzymes. It should be noted that chitinases have been identified as virulence factors in a variety of other studies, but no conclusive evidence has been provided for their function [88,89]. Some studies have suggested that the role of virulence-related chitinases may be degrading substrates other than chitin, e.g., the GlcNAc containing glycans of glycoproteins [90,91]. Such substrates may be present in the salmon skin mucus.

Beside proteases, another group of abundant bacterial proteins identified were related to transport processes, more specifically iron transport via siderophores or heme (Figure 4 and Appendix A; panel C). Iron acquisition is essential for bacteria, since many enzymes are dependent on this metal as a cofactor [92,93,94]. Siderophores and siderophore receptors have also been associated with bacterial pathogenesis, where their role is to acquire iron within their vertebrate hosts (where the level of free iron is low) in order to survive and proliferate [69,95]. Although siderophores cannot be detected by proteomics, their respective outer membrane receptors (ferrichrome receptor; Figure 4, panel C), and iron complex outer membrane receptor protein; Appendix A, panel C) were displayed at low, medium and high abundance from day 0 to day 9. The expression of siderophore transporters and siderophore receptors is associated with virulence in several Vibrio species (e.g., the eel pathogen *Vibrio vulnificus*) [96]. For other bacteria, such as *Pseudomonas aeruginosa*, siderophores also act as signaling compounds for production of virulence factors such as proteases [97]. It is tempting to speculate that the high abundance of both siderophore receptors and proteases observed in the exoproteome may be functionally connected.

An unexpected result was the low amount of bacterial glycoside hydrolases related to mucin degradation, since mucin is thought to represent the most abundant glycoprotein in the mucus [40]; only one GH33 bacterial sialidase was identified (Vibrio; A0A178JAP7). In the current study, the average mucin concentration was ~16 mg/mL (Table 2), thus representing a plentiful nutrient resource. The mucus viscosity declined visibly upon incubation and bacterial growth, indicating mucin depolymerization (results not shown). In separate study, Padra et al showed that the skin mucins of *Salmo salar* are relatively simple, mostly glycosylated by a disaccharide made of *N*-acetylgalactosamine and sialic acid [18,61]. Since the mucin was so abundant in the mucus samples, one would expect to identify multiple family GH33 sialidases or family GH101 endo-α-*N*-acetylgalactosaminidase in the exoproteome. However, rather than glycoside hydrolases, proteases were the dominant hydrolase activity identified. Proteases are known to be important enzymes for mucin depolymerization by bacteria [8] and the resulting glycopeptides may be utilized as nutrients. Indeed, such mucin processing system has been demonstrated for *Capnocytophaga canimorsus,* a bacterium feeding on mucins in dog saliva [98]. Interestingly, the most abundant bacterial proteins identified at day 0 and also in the other time points are proteases and TonB transporters, where the latter may transport glycopeptides into the cells [99]. 

### 4.3. The Salmo salar Skin–Mucus Proteome

In addition to the analysis of bacterial proteins present in the mucus exoproteome, we also analyzed the skin–mucus proteins associated with farmed Atlantic salmon and their putative roles in controlling the bacterial population in the mucus. GO analysis of the *Salmo salar* exoproteome showed the presence of proteins involved in biological regulation processes, transport processes, metabolic processes, cellular processes, and multiorganism processes (Appendix A). It is well known that one of the major roles of the skin mucosal surface is to protect fish towards pathogens [40] and the many proteomic studies on fish skin–mucus have identified a variety of proteins considered to be important for the mucosal immune system. In the present proteomic dataset we detected several types of ribosomal proteins, lysozymes and histones (Appendix A), all known for having antimicrobial properties [100,101]. In addition, structural proteins were detected, which is in agreement with other studies [33,102,103]. Actin has been identified in mammalian mucus mainly during infection processes [104,105], thus its presence in fish may be correlated simply with a discharge of damaged cells into mucus. However, some studies suggest that actin and keratin represent part of the fish immune response and defense [102,106]. Surprisingly, hemoglobin was also detected in our analysis. The presence of hemoglobin in mucus may be correlated with blood contamination during the sampling procedure, although this protein has previously been identified in the epidermal mucus of stressed fish [107], suggesting a possible role in the fish skin mucus. For instance, hemoglobin-derived peptides in vertebrates have been shown to have antimicrobial activity [108]. 

Most of the salmon proteins identified in mucus (85%) were predicted to be intracellular (Table 1). Proteomic studies of the human intestinal mucus showed similar characteristics, where the high abundance of intracellular proteins was explained by the presence of detached epithelial cells present in the mucus that may leak or rupture upon sample preparation [2]. Indeed, epithelial cell shedding is commonly observed in the intestine [109] and is related to maintaining the tissue homeostasis [110]. Although mammalian intestinal mucus and fish skin–mucus are not directly comparable, it is tempting to speculate the detection of epithelial cells in mucus due to lysis during sample collection, or to bacterial degradation, which may cause the leakage of intracellular proteins. A recent study by Rudi et al. [111] showed that intestinal mucus of *Salmo salar* smolts contained a large amount of apoptotic cells resulting in the extraction of host DNA. Other studies of fish skin–mucus also report a dominance of putatively intracellular proteins [33,102] and it is possible that many of these originate from cells found on the skin surface. Unfortunately, there is no appropriate way to avoid collection of epithelia cells together with mucus. In the present study, we attempted to minimize the inclusion of epithelial cell proteins in our samples by collecting the mucus from the fish as gently as possible; first, by putting the fish in sterile plastic bags so the majority of the mucus could be obtained by draining the plastic bag. Second, the remaining mucus on the fish skin was gently removed by using a rounded plastic spatula. Analyses of tissue expression bias of the genes encoding the *Salmo salar* proteins from mucus also supported that that these proteins originated mainly from skin cells (Appendix A).

Since the concentration of mucins is high in the fish skin–mucus (Table 2), one would expect to find mucins in the proteome despite their high degree of glycosylation and low complexity amino acid sequence. Indeed, proteins annotated as mucin-5B-like (A0A1S3PVV9), intestinal mucin like (A0A1S3QCU3), and mucin-5AC-like (A0A1S3KVA9) were detected in relatively large abundances in the proteome. A recent study by Sveen et al. characterized seven mucin encoding genes from Atlantic salmon and their tissue specific expression [19]. Of these seven, two genes were predominantly expressed in the skin (*muc5ac.1*: gene ID XP_013982550.1, protein ID A0A1S3KVA9 and *muc5b*: gene ID XP_014031349.1, protein ID A0A1S3PVV9). Proteins representing both these genes were identified in the present proteome (see text above and Table 1), but none of the other five mucins. This further highlights the tissue specificity of some of the mucins as suggested by Sveen et al. In addition, our data suggest that the “intestinal mucin like” mucin (protein ID A0A1S3QCU3) is expressed in the skin. 

Intriguingly, glycoside hydrolases predicted to deglycosylate mucins were also detected in abundance amongst the *Salmo salar* proteins (several putatively secreted GH33 sialidases, GH20 N-acetylhexosaminidases and GH89 α-N-acetylglucosaminidases; see Appendix A). It is not known why such enzymes are present in the mucus, but they may be involved in remodeling of the mucins or degrading/modifying bacterial lipopolysaccharides; it is established that some bacteria decorate their cell surface with common host glycans like sialic acid in order to evade the immune system [112]. In mammals, it has been shown that infections induce mucin glycosylation changes, which in turn affect pathogen adhesion [6,113]. Similarly, infection promote changes in the mucosal glycosylation in fish [4,5].

## 5. Conclusions

The present study gives a first glance of the skin–mucus proteins of farmed *Salmo salar*, taking into account both proteins associated with the salmon and the bacterial community. The Atlantic salmon skin–mucus exoproteome revealed proteins involved in different processes such as biological regulation processes, transport processes, metabolic processes, cellular processes, and multiorganism processes. These results indicate the occurrence of cell lysis, which represent a general challenge when dealing with the proteome of mucosal surfaces. Nevertheless, multiple antimicrobial proteins and enzymes were detected, emphasizing the protective role of mucus. Moreover, our study indicates that the putative antimicrobial properties of the isolated skin–mucus did not prevent bacterial proliferation using mucus constituents as the main nutrient source, a property monitored over time by determining the temporal proteome dynamics of the most abundant genera. At the beginning of the incubation period, the largest proportion of proteins (>99%) belonged to the host. Successively, proteins associated with the genera Vibrio and Pseudoalteromonas dominated the mucus exoproteome, highlighting a clear decrease of the salmon proteins. Indeed, the GO analysis showed the expression of several types of proteases by these bacteria, which may be used to degrade salmon proteins to grow in mucus. It should be noted that the present data only indicates the potential of mucus as a nutrient source when separated from the fish skin. When the mucus is present on the skin of healthy fish, the mucosal immune system will most likely not allow prominent bacterial proliferation (corroborated by the low amount of bacterial proteins in the non-incubated mucus samples; Figure 2). The use of label-free quantification mass spectrometry to characterize the salmon skin–mucus proteome, focusing on both salmon and bacterial proteins, in addition to their temporal proteome dynamics, represent a new approach of study, which may contribute to better understanding the etiology of skin disorders associated with farmed fish. 

## Figures and Tables

**Figure 1 genes-10-00515-f001:**
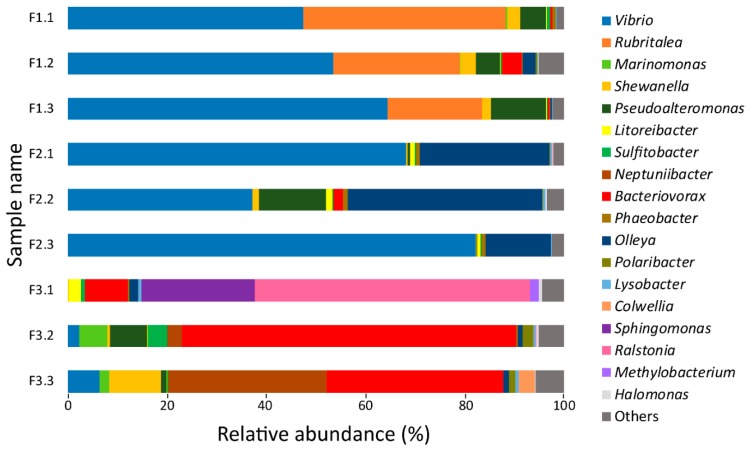
Relative abundances of the bacterial genera identified after 9 days of salmon skin–mucus incubation with sterilized seawater by 16S rRNA gene sequencing analysis. Each technical replicate per sampled fish is visualized separately in this figure. All the OTUs < 0.2% are indicated as “Others”.

**Figure 2 genes-10-00515-f002:**
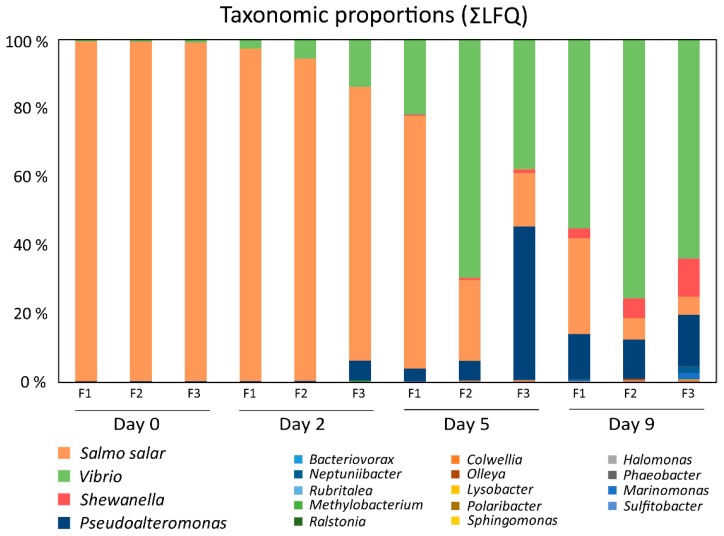
Taxonomic proportions of proteins in mucus over time. The bar chart shows the taxonomic proportions at the genus level, based on expressed proteins, assigned to the bacterial community over 9 days of mucus incubation with sterilized seawater. In addition, the proteins associated with salmon are displayed as “*Salmo salar*”.

**Figure 3 genes-10-00515-f003:**
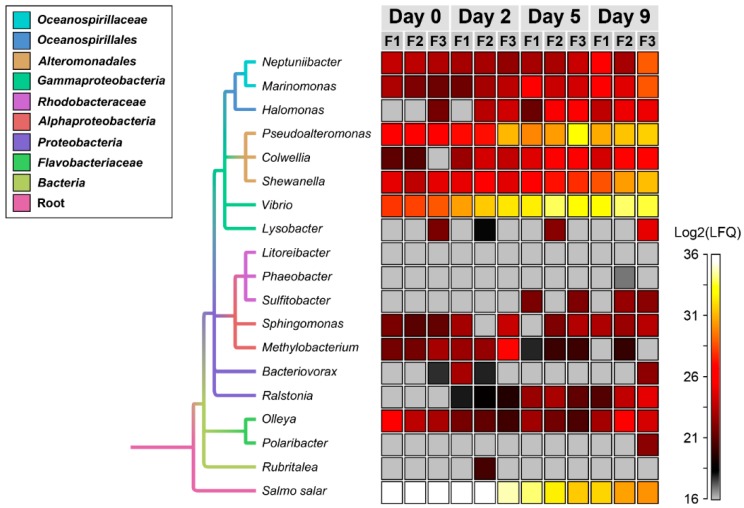
Heat map of the total expressed proteins. The heat map shows the abundances of expressed proteins, summarized at genus level, during 9 days of mucus incubation with sterilized seawater. It ranges from low abundance (gray), to medium abundance (red) and high abundance (white). The genera are sorted based on a phylogenetic tree made with the NCBI Common Tree tool and visualized using the software FigTree version 1.4.

**Figure 4 genes-10-00515-f004:**
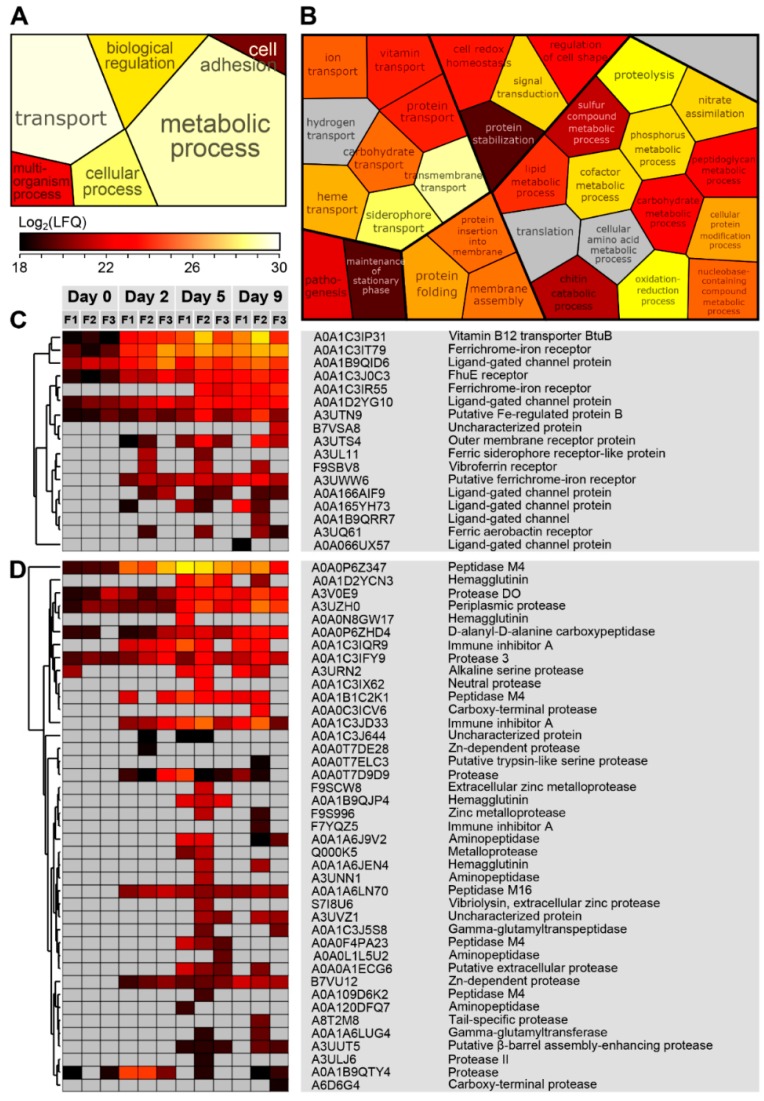
Functional categorization of secreted proteins from the genus Vibrio. The secreted proteins from Vibrio were grouped according to their biological function using Gene Ontology (GO), and the abundance for each group was calculated as the sum of all the individual proteins. High-level GO terms, i.e., generic biological processes, are shown in Panel A and the color of each block reflects the abundance. Panel B shows a similar representation for medium-level GO terms, i.e., more specific biological processes. Gray blocks indicate that no proteins could be mapped to that specific GO-term. Panel C shows a heat map of the proteins associated with ‘siderophore transport’, while Panel D shows proteins assigned to ‘proteolysis’. The heat map ranges from low abundance (black), to medium abundance (red) and high abundance (white).

**Table 1 genes-10-00515-t001:** The total number of proteins (all the time points) assigned to the different genera identified by 16S rRNA sequencing (Figure 1) and the amount of proteins predicted as secreted.

Genus	Protein Count	Secreted ^a^
*Salmo*	3583	523 (15%)
*Vibrio*	2017	390 (19%)
*Pseudoalteromonas*	1621	427 (26%)
*Shewanella*	519	101 (19%)
*Marinomonas*	154	27 (18%)
*Neptuniibacter*	146	24 (16%)
*Colwellia*	33	12 (36%)
*Olleya*	22	10 (45%)
*Halomonas*	18	0
*Sphingomonas*	11	3 (27%)
*Lysobacter*	6	0
*Methylobacterium*	6	2 (33%)
*Ralstonia*	4	0
*Bacteriovorax*	2	0
*Phaeobacter*	1	0
*Polaribacter*	1	0
*Rubritalea*	1	0
*Sulfitobacter*	1	0
*Litoreibacter*	0	0

^a^ Predicted using SignalP.

**Table 2 genes-10-00515-t002:** Mucin concentration estimated in the skin–mucus samples.

**Sample**		**Mucin (mg/mL)**
F4		9.85
F5		20.73
F6		22.18
F7		13.18
**Sample**	**Time (h)**	
F8	0	8.9 (3.6) *
F8	12	9.2 (2.9) *
F8	24	6.2 (1.1) *
F8	48	4.1 (0.4) *

* Values are shown as median (*n* = 3) with range in parenthesis.

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
