# Peer review of "The Farmed Atlantic Salmon (Salmo salar) Skin–Mucus Proteome and Its Nutrient Potential for the Resident Bacterial Community"

_genes, 2019, doi:10.3390/genes10070515_

Reviewer 1 Report

The manuscrit genes-505484             describes a very well designed study. The authors studied the bacterial community of the Salmo salar skin mucus and also identified expressed proteins by using label-free quantitative mass spectrometry.

They incubate skin mucus with sterilized seawater during 9 days to evaluate the capacity of bacterial community to use mucus components to growth in this environment

The dominant genus is Vibrio according to 16S analysis and expressed proteins showing the importance of this genus.

The number of animals (3 by cultural replicate and 9 in total) is normally not enough to compare and analyse skin microbiota regarding the very important inter-individual variability observed in skin microbiota.  But here, we are in a first in vitro exploratory approach, so we can accept this. But it would have be interesting to have the coverage information (at least in supplementary date to know the quality of sequencing. Furthermore it is necessary to publish the sequences (SRA for example). Then, cultural replicate show a large variation in the sample F3 showing the limit of this method.

At the start of incubation, the Salmo salar proteins were dominant, then replaced by bacterial proteins:

This work provide interesting informations regarding the pattern of protein expressed: transport, metabolic and cellular processes. The authors identified also antimicrobial proteins and they conclude that it was not enough to prevent bacterial proliferation. They should conclude carefully, we are not with a  alive fish with all in active immunity defense.

To conclude, I purpose to do modifications regarding my major concerns explained above.

Author Response

Please see the attached document for a point-by-point response.

Reviewer 2 Report

The manuscript genes-505484 shows a novel approach and very interesting results to understand the dynamic between microbiota and mucus in salmonids.

Why the mucus were mixed? (bag and skin mucus). Please explain the idea behind this approach and how this impact the conclusions.

Why only 100ul were considered for microbiota analyses?

Comments:

M&M: Please explain criteria for protein match. cutoff? range? domain?

Table 1. why Litoreibacter, the score is 0.

Line 247: 3.1.1 is confusing 16S.

Author Response

(The authors gave the same response as above.)
